# ONLINE ALGORITHM CONFIGURATION FOR MILP RE-OPTIMIZATION WITH LLM GUIDANCE

## ABSTRACT

In this work, we study the re-optimization setting for mixed-integer linear programs, where solving sequentially related instances can benefit from both adaptive solver parameter configuration and the reuse of historical information from previous solves. However, modern solvers expose hundreds of tunable parameters, yielding a large configuration space; and the effectiveness of re-optimization techniques (e.g., warm starts or branching statistics) varies substantially across problem families. To address these challenges, we formulate a generalized algorithm configuration problem that jointly determines solver built-in parameters and the selective use of historical information within a reduced configuration space. Given the sequential nature of re-optimization and the limited number of available instances, offline methods that require large datasets are impractical, so adaptive online configuration selection becomes essential. We therefore propose a two-stage framework: (i) configuration space reduction via large language models, which generate a compact portfolio of candidate configurations; and (ii) adaptive online selection using multi-armed bandit algorithms to minimize solving cost over the sequence. Empirical results on the MIP Workshop 2023 re-optimization benchmarks demonstrate that our method substantially outperforms default SCIP and Gurobi configurations as well as strong baselines, achieving solving time reductions of up to $54.18\%$, without requiring prior validation data or supervised training.

## 1 INTRODUCTION

Many real-world decision problems are naturally modeled as mixed-integer programs (MIPs). When the underlying system evolves over time, as in hybrid model predictive control Richards & How (2005); Frick et al. (2019); Marcucci & Tedrake (2020), dynamic vehicle routing Dondo & Cerdá (2006); Ozbaygin & Savelsbergh (2019); Andersen et al. (2024), unit commitment in power systems Morales-España et al. (2013); Gentile et al. (2017); He et al. (2024), and dynamic production planning Wolsey (1997); Cedillo-Robles et al. (2020); Dunke & Nickel (2023), practitioners rarely solve a single static instance. Instead, they face a *sequence* of closely related instances that share a common structure but differ in parameters such as objective coefficients, constraint right-hand sides, or variable bounds. We refer to this sequential setting as *re-optimization*; in this paper we focus on the mixed-integer linear programs (MILPs) case.

Instead of solving each instance from scratch, in the setting of re-optimization one can exploit information from previous instances in three complementary ways: (i) warm starts from historical solutions Berthold (2006); Gamrath et al. (2015); (ii) reuse branch-and-bound (B&B) information such as pseudocosts or conflict statistics Achterberg et al. (2005); Patel (2024a); (iii) automated parameter tuning Xu et al. (2011). Although these approaches have been empirically effective, several research gaps remain. First, their effectiveness is highly problem specific, simply enabling all re-optimization mechanisms may not yield the best performance Patel (2024b). Second, many search-based or learning-based parameter tuning methods require solving large numbers of instances offline to train predictors or validate configurations, which is impractical in re-optimization scenarios where only limited data are available and system dynamics evolve over time. Third, the parameter space of modern MILP solvers is already very large, and re-optimization introduces additional tunable options, further exacerbating the challenge of efficient exploration.

These limitations motivate us to propose a generalized algorithm configuration framework for re-optimization. To address the first gap, we treat the use of re-optimization mechanisms (e.g., warm starts) as tunable options alongside built-in solver parameters. To address the second, we design a lightweight online method based on multi-armed bandits (MAB), which avoids costly offline training or searching and adapts to dynamic changes. To address the third, we leverage the zero-shot capabilities of large language models (LLMs) to reduce the algorithm configuration space, generating a compact but diverse configuration candidate set from which the bandit algorithm can adaptively select. Our main contributions are as follows:

- We formally define the MILP re-optimization problem, where a solver faces a sequence of structurally related instances and can leverage a finite history window of past solutions and solver information. Within this setting, we cast the task as a generalized algorithm configuration problem that jointly considers built-in solver parameters and re-optimization techniques.

- We develop a lightweight two-stage approach that first introduces LLMs to generate configuration candidates, thereby reducing the combinatorially large algorithm configuration space to a compact candidate set, and then designs an MAB-based strategy for adaptive online configuration selection.

- We demonstrate on challenging benchmarks with multiple varying components and large-scale instances that our method consistently outperforms default SCIP and Gurobi solvers as well as strong baselines, achieving solving-time reductions of up to $54.18\%$, without requiring offline training or validation data.

## 2 LITERATURE REVIEW

**Re-optimization for MILPs.** Re-optimization of sequentially related MILP instances has been investigated extensively, with prior work exploring techniques that leverage historical information from previously solved similar instances to accelerate solving new ones. Modern solvers such as Gurobi Gurobi Optimization (2023) and SCIP Bestuzheva et al. (2021) include warm-start features, whereby feasible or incumbent solutions from past instances are provided as primal hints. Gamrath et al. (2015) presents a branch-and-bound scheme in SCIP that reuses the search frontier from a solved instance, mainly focusing on objective coefficient variations. More recently, Bolusani et al. (2024) delivers extensive benchmarks covering variations in objective coefficients, variable bounds, right-hand sides, and constraint matrix coefficients. Patel (2024b) achieves leading performance on these benchmarks by combining reuse of primal solutions, branching history, and automated parameter tuning to adapt across multiple instances. Zhang et al. proposes a two-stage re-optimization framework: first predicting a high-confidence solution subspace from historical solving trajectories, then partially fixing variables within this subspace using Thompson Sampling to accelerate the search. The method is mainly effective for quickly identifying feasible solutions without necessarily reaching optimality, and it does not extend to re-optimization scenarios with multiple varying components.

**Algorithm configuration.** Modern MILP solvers such as SCIP expose hundreds of tunable parameters across many components (e.g., branching rules, cut separators, presolve options, heuristics, conflict analysis, symmetry handling) Bestuzheva et al. (2021). In addition, the use of re-optimization methods introduces further tunable choices. For example, when leveraging primal hints from previously solved instances, one may decide how many incumbent solutions to carry forward and from which instances, as well as whether to include only integer variables or all variables Patel (2024b). These solver-native and re-optimization parameters yield a combinatorially large configuration space. The task of selecting suitable parameter values is typically framed as an *algorithm configuration* problem, where extensive offline search is conducted to identify configurations that perform well across a distribution of instances Hutter et al. (2007; 2009); Xu et al. (2011); Eryoldaş & Durmuşoglu (2022). However, such approaches are computationally prohibitive in practice, as they require evaluating various configurations on large training sets.

Learning-based methods have been proposed to predict good configurations or guide the search over large configuration spaces Biedenkapp et al. (2020); Adriaensen et al. (2022); Valentin et al. (2022); Li et al. (2023); Hosny & Reda (2024). Many of these approaches focus on *instance-wise* algorithm

configuration, where handcrafted or learned features are mapped to solver configurations. While effective, such methods typically depend on costly feature engineering, supervised training, and large datasets to ensure generalization. In the re-optimization setting, Patel (2024b) addresses algorithm configuration by applying an MAB framework. Instead of time-consuming offline training, they adaptively tune parameters online. However, their approach treats each parameter independently as a separate bandit problem, which hampers convergence and forces them to restrict tuning to only a small subset of parameters, limiting achievable performance. These observations motivate our question: *can we design a more lightweight, training-free approach that remains effective in re-optimization scenarios?*

Lawless et al. (2025) recently proposed an LLM-based method for separator configuration with minimal training data, leveraging instance descriptions and formulations. Yet, LLMs may hallucinate and generate inconsistent configurations, necessitating an additional validation set—impractical in re-optimization scenarios with limited instances. Furthermore, their scope is restricted to separators, whereas re-optimization exposes richer historical information and a broader range of tunable parameters. These limitations motivate our work on generalized re-optimization algorithm configuration, leveraging the zero-shot capabilities of LLMs for space reduction and online bandit-based adaptation for selection.

## 3 Problem Formulation

In this section, we first present formal formulations of the MILP model and the re-optimization setting. We then define the generalized algorithm configuration problem. Finally, we describe two sub-tasks: (i) generating configuration portfolios to reduce the configuration space, and (ii) performing online configuration selection over these configuration candidates.

**Mixed Integer Linear Programming (MILP).** An MILP problem can be represented as follows:

$$\min_{\boldsymbol{x}\in\mathbb{R}^n} \boldsymbol{c}^\top \boldsymbol{x}, \quad \text{s.t. } \boldsymbol{x}\in\mathbb{X}_{\text{MILP}}=\left\{\boldsymbol{x}:\boldsymbol{A}\boldsymbol{x}\circ\boldsymbol{b}, \quad \boldsymbol{l}\leq\boldsymbol{x}\leq\boldsymbol{u}, \quad x_j\in\mathbb{Z},\forall j\in\mathbb{I}\right\}, \quad (1)$$

where $\boldsymbol{A}\in\mathbb{Q}^{m\times n}$, $\boldsymbol{c}\in\mathbb{Q}^n$, and $\boldsymbol{b}\in\mathbb{Q}^m$. The symbols $\boldsymbol{l}$ and $\boldsymbol{u}$ denote the lower and upper bounds of the variables, respectively, and each component satisfies $l_j\in\mathbb{Q}\cup\{-\infty\}$ and $u_j\in\mathbb{Q}\cup\{+\infty\}$. The relational operator $\circ$ indicates the type of constraint applied to each row, with entries $\circ_j\in\{\leq,=,\geq\}$. The index set $\mathbb{I}\subset\{1,2,\ldots,n\}$ identifies which variables $x_j$ are integer variables.

**Re-optimization problem.** We consider solving a sequence of MILP instances $\{P_t\}_{t=1}^T\subseteq\mathcal{P}$ arising from the same problem family. The instances in each series have a fixed overall problem structure: the number of constraints, and the number, order, and meaning of variables remain the same across the instances in a series. Some or all of the following input data may vary over $t$: objective function coefficients $\boldsymbol{c}$, variable bounds $\boldsymbol{l}$ and/or $\boldsymbol{u}$, constraint right-hand sides $\boldsymbol{b}$, and coefficients of the constraint matrix $\boldsymbol{A}$ Bolusani et al. (2024). At round $t$, the solver faces the current instance $P_t$ and may access a finite history window of length $\tau$:

$$\mathcal{P}_t^{(\tau)}=(P_{t-\tau},\ldots,P_t), \quad H_t^{(\tau)}=(h_{t-\tau},\ldots,h_{t-1}),$$

where $h$ denotes information obtained from the solver, such as primal solutions, branching information, or cut statistics. After solving the MILP problem, the solver produces an output $z_t=f(\mathcal{P}_t^{(\tau)},H_t^{(\tau)})$, where $f$ denotes the (black-box) solver execution. The output $z_t$ captures relevant performance signals, such as solving time, primal–dual integral, optimality gap, or objective value. The cost is then defined as a functional of this output, $C(z_t)$, which provides a scalar performance measure for round $t$. The re-optimization problem is to design a policy that minimizes the cumulative cost over the sequence:

$$\min\sum_{t=1}^T C(z_t).$$

**Algorithm configuration.** In our re-optimization setting, MILP solvers expose numerous tunable *built-in parameters*. In addition, our framework treats optional *re-optimization parameters* as part of the configuration space, controlling whether and how prior information is reused. The algorithm configuration problem is to derive a policy that automatically selects these parameters to accelerate

the solution for each forthcoming instance in a series. Let $\mathcal{M} = \{1, \ldots, M\}$ index the set of configurable parameters. For each $i \in \mathcal{M}$, let $\mathcal{S}_i$ denote the finite set of admissible choices with cardinality $|\mathcal{S}_i| = K_i$. For instance, one configurable parameter may be whether to use primal hints from previous instances as a warm start in re-optimization ($K_i = 2$, with 0 indicating not used and 1 indicating used). Another parameter corresponds to separator aggressiveness as implemented in commercial solvers such as Gurobi ($K_i = 3$, with 0 = off, 1 = moderate, and 2 = aggressive). The overall configuration space is the Cartesian product $\mathcal{S} = \mathcal{S}_1 \times \cdots \times \mathcal{S}_M, |\mathcal{S}| = \prod_{i=1}^{M} K_i$, and a configuration is an element $s = (s_1, \ldots, s_M) \in \mathcal{S}$ with $s_i \in \mathcal{S}_i$ specifying the choice for algorithm parameter $i$. Note that $s \in \mathcal{S}$ does not correspond to a single parameter choice, but rather to a complete configuration across all parameters. For example, if $M = 2$ with parameters (Primal Hint, CoverCuts), then $s = (1, 2)$ denotes a configuration in which the primal hint is enabled and the cover-cut separator is set to aggressive.

In the re-optimization setting, combining algorithmic choices (e.g., whether to use warm starts or exploit historical branching information) with built-in solver parameters (e.g., separator aggressiveness) yields a combinatorially large configuration space, making direct exploration impractical. We therefore formulate a configuration space reduction task as follows.

**Configuration space reduction.** To reduce the configuration space, given a problem description $\varphi$ (e.g., instance features or textual context), we seek a subset $\mathcal{S}' \subseteq \mathcal{S}$ that still contains configurations with strong performance. In this work, LLMs are used to generate a manageable portfolio of candidate configurations. A multi-armed bandit algorithm then adaptively selects among these candidates, where each configuration corresponds to an arm.

We formalize online configuration selection over the candidate portfolio as a regret–minimization problem:

**Policy and regret.** A policy $\pi$ selects configurations adaptively from the reduced space: $s_t = \pi(\mathcal{P}_t^{(\tau)}, H_t^{(\tau)}) \in \mathcal{S}'$. Applying configuration $s_t$ yields an output $z_t = (\mathcal{P}_t^{(\tau)}, H_t^{(\tau)}; s_t)$ and corresponding cost $C(z_t)$ as defined in the re-optimization problem. To align with the bandit literature, we define the reward as the negative cost: $r_t(s_t) = -C\big(f(\mathcal{P}_t^{(\tau)}, H_t^{(\tau)}; s_t)\big)$. Since $(\mathcal{P}_t^{(\tau)}, H_t^{(\tau)})$ are fixed at round $t$, we abbreviate the reward as $r_t(s_t)$. Performance is then measured by cumulative regret:

$$\mathcal{R}_T(\pi) = \max_{s \in \mathcal{S}'} \sum_{t=1}^{T} \mathbb{E}[r_t(s)] - \sum_{t=1}^{T} \mathbb{E}[r_t(s_t)].$$

This quantifies the gap between the policy $\pi$ and the best fixed configuration in hindsight.

## 4 METHODOLOGY

In this section, we present a two-stage approach for sequentially related MILP instances. First, an LLM proposes configuration candidates $s \in \mathcal{S}$; querying it $N$ times yields a reduced portfolio $\mathcal{S}' \subseteq \mathcal{S}$ with $|\mathcal{S}'| = N$. Second, an MAB algorithm adaptively selects configurations from $\mathcal{S}'$ along the re-optimization sequence. Figure 1 depicts the workflow.

### 4.1 OFFLINE CONFIGURATION SPACE REDUCTION WITH LLMS

Inspired by Lawless et al. (2025), we provide detailed descriptions of the re-optimization problem series and the tunable parameters, so as to supply sufficient context for LLMs to generate reasonable and diverse configurations. In what follows, we introduce the specific design choices made for the re-optimization setting. Details of the LLM prompts are provided in Appendix A.3.

**Problem description.** The re-optimization datasets Bolusani et al. (2024) used in this work provide comprehensive context, including provenance, series-level metadata, and the components that vary across instances. We convert this information into a structured prompt for each problem series with three blocks: (i) *Provenance & formulation.* We provide a concise natural-language description along with the canonical IP/MIP formulation (sets, variables, and constraints). For example, explicitly describing capacity constraints can guide the LLM to enable separators such as implied bound cuts. (ii) *Series metadata.* This block specifies the problem size (number of variables and constraints), the binary/continuous variable mix, and, when available, additional structural hints

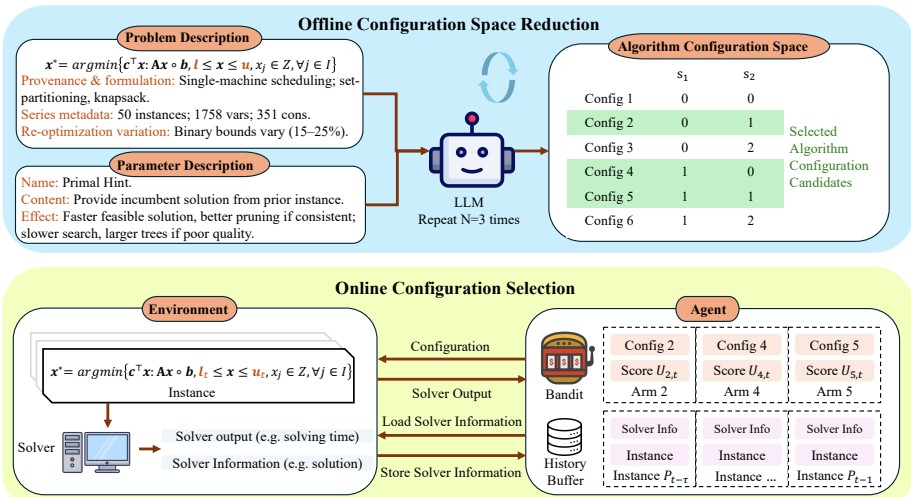

Figure 1: Two-stage framework for MILP re-optimization. Top: Offline configuration space reduction with LLMs. Given problem and parameter prompts, the LLM is queried $N$ times to generate compact portfolios. Bottom: online configuration selection via MABs. At each round $t$, the agent should select a configuration for the current instance $P_t$, and the solver loads historical information. After solving the instance, the environment returns solver output (e.g., solving time) for the agent to update all arm scores and store solver information for future rounds.

such as sparsity or density. (iii) *Re-optimization variation.* We summarize which components vary across the series, such as objective coefficients, right-hand sides, or variable bounds. For instance, objective drift often encourages warm starts, whereas bound changes may reduce the reliability of primal hints. Each block provides signals that help the LLM generate configurations aligned with the structural and dynamic properties of the problem series.

**Parameter description.** For each solver, we list the tunable parameters together with their discrete choices. We provide both the solver-specific names and a detailed description of each parameter. These descriptions are consolidated from multiple sources, including research papers, textbooks, solver documentation, and empirical insights. Since the parameter choices typically control not only whether a function is enabled or disabled, but also to what extent, we explicitly summarize the strengths and limitations of each parameter to guide LLMs in deciding both whether and how aggressively to activate a function.

For example, Primal Hint (MIP warm-start) supplies an incumbent solution from previous instances. This can significantly reduce time to derive a feasible solution at the early stage and improve pruning efficiency when the hint is of high quality and structurally consistent across instances. However, if the incumbent solution has a poor objective value, lower bounds cannot prune effectively, leading to larger search trees. Another tunable parameter, Root-only cuts, controls whether cutting planes are generated exclusively at the root node. On the one hand, enabling root-only can reduce the overhead of cut separation and LP solves at non-root nodes. On the other hand, it may fail to sufficiently strengthen the dual bound, thereby decreasing pruning opportunities and potentially increasing the number of explored nodes. Describing such strengths and limitations is crucial for LLM to generate algorithm configurations: it enables the model to weigh trade-offs (e.g., speed versus pruning power) and to adapt recommendations more effectively in the re-optimization setting, where historical context can amplify both the benefits and risks of these parameter choices.

### 4.2 ONLINE CONFIGURATION SELECTION VIA MULTI-ARMED BANDITS

Through offline configuration space reduction with LLMs, we derive a reduced candidate set $\mathcal{S}' \subseteq \mathcal{S}$. We then formulate the task of adaptively selecting configurations from $\mathcal{S}'$ as an MAB problem: at each round $t$, the policy $\pi$ chooses a configuration $s_t \in \mathcal{S}'$, observes the resulting reward $r_t(s_t)$, and updates its decision rule based on past observations. The goal is to minimize the cumulative cost relative to the best configuration in hindsight.

We instantiate the configuration selection policy $\pi$ using the Upper Confidence Bound algorithm with a tunable exploration coefficient $\alpha > 0$ to scale the bonus term (Auer et al., 2002). For each candidate configuration $s \in \mathcal{S}'$, let $\hat{\mu}_s(t)$ denote its empirical mean reward up to round $t$, and let $n_s(t)$ denote the number of times $s$ has been selected. UCB1 computes the score

$$U_s(t) \;=\; \hat{\mu}_s(t) \;+\; \alpha\sqrt{\frac{2\ln t}{n_s(t)}}, \tag{2}$$

and selects the configuration $s_t = \arg\max_{s \in \mathcal{S}'} U_s(t)$. The first term promotes exploitation of configurations with high empirical rewards, while the second term is an exploration bonus that decays with the number of pulls. More details of the MAB algorithm are provided in Appendix A.4.

## 5 EXPERIMENT

In this section, we evaluate our proposed framework against default solver configurations and representative baselines. Our experiments are designed to address the following research questions: (i) *Comparative performance.* How does our framework perform relative to default solver settings and baselines? (ii) *Offline configuration space reduction.* Can LLMs efficiently reduce the algorithm configuration space by generating reasonable and diverse candidate configurations? (iii) *Online configuration selection.* Can bandit algorithms adaptively select effective configurations over the sequential instances within a re-optimization series?

### 5.1 EXPERIMENT SETUP

All experiments are conducted on a high-performance computing cluster with Intel Xeon Platinum 8628 CPUs. All Gurobi experiments use Gurobi 12.0.3, and all SCIP experiments use SCIP 9.02. We set the number of candidate configurations to $N = 5$ for each re-optimization problem series. However, since single-shot LLM outputs may be noisy or redundant, and following prior work Lawless et al. (2025), we deliberately over-generate a larger pool of 100 configurations and then apply $k$-medoids clustering to condense them into $N = 5$ representative candidates. The exploration coefficient and the length of history window are fixed at $\alpha = 1$ and $\tau = 5$, respectively. We further investigate the sensitivity of these parameter choices in this section.

**Benchmarks.** We evaluate our proposed method and baselines on the datasets provided by The MIP Workshop 2023 Computational Competition ON Re-optimization Bolusani et al. (2024). We select 10 datasets from this benchmark, each corresponding to a re-optimization problem series consisting 50 sequential instances. To analyze performance across different difficulty levels, we categorize the datasets into easy and hard groups. *Easy datasets* involve only a single varying (e.g., objective coefficients), and have relatively small problem sizes, with no more than 1,457 integer variables. *Hard datasets* are characterized either by having at least two (up to four) varying components, by significantly larger problem sizes (up to 63,009 integer variables), or by both. For each dataset, we further split the 50 instances into 45 evaluation instances and 5 validation instances. Note that validation instances are only used by one baseline method; our approach and all other baselines do not require validation data. More details of the benchmarks are provided in Appendix A.2.

**Evaluation metrics.** We consider the average *solving time* across a re-optimization sequence as the primary evaluation metric: Time $= \frac{1}{T}\sum_{t=1}^{T} z(t)$, where $z(t)$ denotes the solving time of instance $t$. We also report the *relative improvement* over default solvers: Improve $= \frac{\text{Time}_{\text{default}} - \text{Time}_{\text{method}}}{\text{Time}_{\text{default}}}$. All instances are solved to optimality, subject to a maximum time limit of 400 seconds. If an instance does not terminate within the limit, its solving time is set to the maximum, ensuring fairness and preventing a few particularly hard instances from dominating the aggregate evaluation.

**Baselines.** We compare against four baselines: (i) *Default solvers (Default).* SCIP and Gurobi with default configurations. (ii) *Progressively Tuning (Tuning)* Patel (2024b). The winning solution of the MIP Workshop 2023 re-optimization competition. Each parameter (primal hints, root cuts, non-root cuts) is modeled as an independent bandit problem, with arms corresponding to its discrete choices (e.g., on/off). Implemented only for SCIP, as Gurobi does not expose root/non-root separation controls. (iii) *LLM with validation (LLM-validation)* Lawless et al. (2025). LLMs generate separator configurations, followed by validation-set selection of the best candidate. This

constitutes a strong baseline, as it leverages additional validation instances to select the best configuration among the LLM-generated candidates. (iv) *LLM-cold-start.* LLM-generated configurations are clustered using $k$-medoids, and the representative of the largest cluster is selected directly for evaluation. Appendix A.1 provides additional details of the experimental setup.

## 5.2 EXPERIMENT RESULTS

Table 1: Comparison of different methods across 10 re-optimization benchmarks with SCIP solver. Each method is evaluated by solving time (s) and relative improvement compare to Default SCIP solver (%). The first 5 benchmarks are categorized as *easy*, and the last 5 as *hard*.

| | Default | Tuning | | LLM-validation | | LLM-cold-start | | LLM-MAB (Ours) | |
|---|---|---|---|---|---|---|---|---|---|
| Dataset | Time ↓ | Time ↓ | Improv. ↑ | Time ↓ | Improv. ↑ | Time ↓ | Improv. ↑ | Time ↓ | Improv. ↑ |
| *Easy datasets* | | | | | | | | | |
| bnd_2 | 252.41(6.70) | 244.66(6.33) | 3.05 | 187.12(9.61) | 25.77 | 235.97(5.67) | 6.48 | **185.01(5.65)** | **26.64** |
| bnd_3 | 355.04(3.81) | 365.83(3.31) | -3.04 | 349.02(17.03) | 2.24 | 358.81(2.41) | -0.79 | **343.79(1.10)** | **3.16** |
| rhs_2 | 80.32(1.80) | 62.25(1.04) | 22.46 | 59.33(2.85) | 26.06 | **47.04(0.46)** | **41.42** | 48.53(0.98) | 39.58 |
| rhs_4 | 77.04(0.59) | 61.06(0.44) | 20.73 | 59.01(2.05) | 23.4 | 47.95(0.34) | 37.76 | **47.53(1.58)** | **38.83** |
| obj_1 | 306.71(8.83) | 241.23(6.22) | 21.32 | 236.28(2.79) | 22.93 | 232.50(1.55) | 24.16 | **229.50(8.83)** | **25.10** |
| *Hard datasets* | | | | | | | | | |
| rhs_3 | 311.94(15.96) | 291.27(13.86) | 6.32 | 296.81(36.49) | 4.35 | 289.24(39.02) | 6.68 | **285.23(8.30)** | **8.30** |
| obj_3 | 145.74(8.78) | 74.30(6.75) | 48.73 | 74.25(10.88) | 48.65 | 67.40(5.42) | 53.60 | **66.63(3.62)** | **54.18** |
| rhs_obj | 397.14(1.09) | 394.92(4.81) | 0.56 | 391.46(4.95) | 1.43 | 388.01(2.91) | 1.95 | **387.99(1.69)** | **1.99** |
| mat | 368.77(4.73) | 361.30(9.34) | 2.00 | 336.55(8.86) | 8.71 | 337.68(9.77) | 8.42 | **335.42(9.50)** | **9.02** |
| all | 116.13(17.78) | 86.78(12.52) | 23.28 | 86.32(4.63) | 24.03 | 86.24(5.95) | 19.35 | **84.60(3.34)** | **25.99** |

Table 2: Comparison of different methods across 10 re-optimization benchmarks with Gurobi solver. Each method is evaluated by solving time (s) and relative improvement compare to Default Gurobi solver(%). The first 5 benchmarks are categorized as *easy*, and the last 5 as *hard*.

| | Default | LLM-validation | | LLM-cold-start | | LLM-MAB (Ours) | |
|---|---|---|---|---|---|---|---|
| Dataset | Time ↓ | Time ↓ | Improv. ↑ | Time ↓ | Improv. ↑ | Time ↓ | Improv. ↑ |
| *Easy datasets* | | | | | | | |
| bnd_2 | 341.76(4.30) | 336.55(6.92) | 1.24(1.19) | 400(0.00) | -17.62(0.66) | **333.38(8.87)** | **2.18(1.77)** |
| bnd_3 | 341.6(4.07) | 330.75(1.29) | 3.17(1.53) | 400(0.00) | -17.11(1.39) | **324.65(3.47)** | **4.95(2.15)** |
| rhs_2 | 18.41(0.28) | 18.25(0.03) | 0.85(1.64) | 19.39(0.02) | -5.08(1.74) | **18.14(0.06)** | **1.41(2.37)** |
| rhs_4 | 18.26(0.18) | 65.39(41.20) | 0.35(3.55) | 17.35(0.23) | 4.98(1.19) | **17.30(0.39)** | **5.25(1.76)** |
| obj_1 | 89.24(1.27) | 89.14(3.17) | 0.08(4.97) | **86.38(0.39)** | **0.73(4.62)** | 88.30(0.74) | -2.23(6.88) |
| *Hard datasets* | | | | | | | |
| rhs_3 | 33.03(4.12) | 30.98(6.66) | 4.54(28.67) | 32.81(4.93) | -0.43(19.92) | **30.44(7.39)** | **6.65(28.18)** |
| obj_3 | 14.38(0.33) | **7.01(0.27)** | **51.30(2.12)** | 12.73(0.36) | 11.42(4.12) | 8.52(0.07) | 40.73(0.88) |
| rhs_obj | 144.93(2.82) | 158.79(24.32) | -9.42(15.34) | 160.42(11.06) | -10.76(8.82) | **143.90(2.37)** | **1.52(1.45)** |
| mat | 44.52(1.55) | 38.99(2.71) | 12.44(5.15) | 36.99(0.45) | **16.77(5.11)** | 36.82(0.66) | 15.81(3.66) |
| all | 8.09(0.44) | 8.20(0.77) | -1.62(10.91) | 8.12(0.22) | -0.48(3.27) | **8.05(0.19)** | **0.30(2.84)** |

**Overall performance comparison.** Table 1 and Table 2 summarize the results on SCIP and Gurobi, respectively, reporting mean solving time with standard deviation as well as relative improvement over the default solvers. Our proposed method *LLM-MAB* achieves substantial speedups compared to default SCIP across all benchmarks, outperforming existing baselines on 9 out of 10 benchmarks and remaining competitive with *LLM-cold-start* on rhs_2. On 6 benchmarks, *LLM-MAB* delivers more than $25\%$ relative improvement, reaching up to $54.18\%$ on obj_3, a hard benchmark with 9599 integer variables. Notably, on the challenging all benchmark—where objective coefficients, bounds, right-hand sides, and matrix coefficients all vary—*LLM-MAB* still achieves a $25.99\%$ improvement, demonstrating its robustness under the most complex re-optimization scenarios. *LLM-MAB* also reduces the solving time of Gurobi across all benchmarks, outperforming

other baselines on 8 out of 10 datasets. On the 5 hard benchmarks, it achieves up to $40.73\%$ relative improvement, highlighting its effectiveness under challenging re-optimization scenarios.

*LLM-validation* serves as a strong baseline since it leverages additional validation instances to identify a single well-performing configuration. We observe that this strategy is particularly effective when one configuration generalizes well across all instances in a series. For example, on obj_3 with Gurobi, it achieves a $51.3\%$ relative improvement by selecting the best configuration from the validation set. In contrast, although *LLM-MAB* attains $40.73\%$ improvement on average, its use of UCB inherently encourages exploration of multiple configurations in the early rounds, which incurs overhead and slightly degrades the total average solving time. This effect is amplified in the re-optimization benchmarks where each series contains only 50 instances (45 for testing), so the cost of early exploration constitutes a larger fraction of the total runtime.

We observe that on certain datasets the solving time budget of 400 seconds is insufficient, as many instances could not reach optimality within the limit. To ensure a fairer evaluation, we repeat the experiments with an extended time limit of 600 seconds in Appendix A.5.

Table 3: Ablation of *LLM-MAB* without general solver parameters (i.e., restricted to separator configurations only). Results are reported as relative solving time improvement (%) over Default SCIP across 10 benchmarks. (Higher is better.)

| Method | bnd_2 | bnd_3 | rhs_2 | rhs_4 | obj_1 | rhs_3 | obj_3 | rhs_obj | mat | all |
|---|---|---|---|---|---|---|---|---|---|---|
| LLM-MAB (Ours) | **26.64** | **3.16** | **39.58** | **38.83** | **25.10** | **8.30** | **54.18** | **1.99** | **9.02** | **25.99** |
| | (3.89) | (0.77) | (0.77) | (2.38) | (4.39) | (8.12) | (3.46) | (0.28) | (3.37) | (11.48) |
| LLM-MAB (Separators only) | 17.48 | 1.78 | 36.07 | 34.42 | 22.92 | 6.64 | 53.50 | 0.45 | 8.55 | 15.64 |
| | (2.66) | (1.51) | (2.38) | (2.04) | (1.96) | (10.17) | (3.58) | (1.03) | (3.03) | (16.25) |

Table 4: Ablation of *LLM-MAB* without online bandit adaptation (i.e., using validation-only selection of a fixed configuration). Results are reported as relative solving time improvement (%) over Default SCIP across 10 benchmarks. (Higher is better.)

| Method | bnd_2 | bnd_3 | rhs_2 | rhs_4 | obj_1 | rhs_3 | obj_3 | rhs_obj | mat | all |
|---|---|---|---|---|---|---|---|---|---|---|
| LLM-MAB (Ours) | **26.64** | **3.16** | **39.58** | **38.83** | **25.10** | 8.30 | **54.18** | **1.99** | 9.02 | **25.99** |
| | (3.89) | (0.77) | (0.77) | (2.38) | (4.39) | (8.12) | (3.46) | (0.28) | (3.37) | (11.48) |
| LLM-Validation (General) | 18.42 | -2.31 | 38.96 | 35.35 | 23.40 | **11.23** | 51.81 | 1.71 | **10.135** | 15.94 |
| | (3.34) | (2.87) | (2.07) | (0.61) | (1.46) | (2.27) | (4.66) | (0.66) | (0.50) | (16.22) |

**Effectiveness of algorithm configuration generation.** We analyze this effectiveness from two aspects. First, we evaluate the extension from separator-only tuning (as in prior work Lawless et al. (2025)) to a more general re-optimization algorithm configuration problem that incorporates both solver-native parameters and historical information. Table 3 reports the ablation study comparing *LLM-MAB* with its separator-only variant, which is studied in recent papers Lawless et al. (2025); Li et al. (2023). The results show that our full configuration space consistently outperforms the separator-only setting across all benchmarks. While the margins vary by dataset, improvements are observed on both easy benchmarks (e.g., $+9.16\%$ on bnd_2) and hard ones (e.g., $+10.35\%$ on all), demonstrating that extending the configuration scope beyond separators yields tangible gains in more challenging re-optimization scenarios.

As mentioned in Section 3, the combinatorially large algorithm configuration space makes direct online bandit-based selection difficult to converge, and therefore necessitates selecting a small set of candidate configurations in advance. The comparison with *Tuning* Patel (2024b) in Table 1 also illustrates this: tuning all parameters independently with a shared reward and without space reduction is ineffective, so they restricted to a few parameters, which still led to degraded results. In contrast, *LLM-cold-start* achieves strong performance—surpassing default SCIP on 9 of 10 benchmarks and default Gurobi on 4 of 10—simply by committing to a single configuration selected via clustering. This highlights that LLMs are capable of producing high-quality solver configurations.

**Effectiveness of online configuration selection.** To assess the benefit of MAB-based online configuration selection in *LLM-MAB*, we include the ablation *LLM-Validation (General)* in Table 4. In

this variant, the configuration space is identical to that of *LLM-MAB*, but the best algorithm configuration is chosen once from the validation set and then fixed for evaluation, without online adaptation. The results show that *LLM-MAB* surpasses *LLM-Validation (General)* on 8 of 10 benchmarks, achieving up to $+10.06\%$ additional improvement on `all`. This demonstrates that bandit-based online selection provides robustness by adaptively exploiting different configurations throughout the series, rather than committing prematurely to a single choice. We also note that when one configuration candidate is clearly superior (e.g., `mat`), *LLM-Validation (General)* may perform slightly better, as it can directly identify this configuration from validation, while *LLM-MAB* incurs exploration overhead before converging.

Table 5: Sensitivity analysis of the exploration coefficient $\alpha$ in the UCB scoring rule. We report relative solving time improvement (%) over Default SCIP across 10 re-optimization benchmarks.

| Method | bnd_2 | bnd_3 | rhs_2 | rhs_4 | obj_1 | rhs_3 | obj_3 | rhs_obj | mat | all |
|---|---|---|---|---|---|---|---|---|---|---|
| LLM-MAB ($\alpha = 0.5$) | 17.69 | 1.72 | **50.37** | 33.82 | 20.23 | 6.43 | 49.81 | 1.64 | 9.09 | 18.67 |
| | (5.48) | (1.07) | (25.71) | (0.59) | (3.61) | (6.43) | (3.80) | (0.51) | (3.80) | (18.26) |
| LLM-MAB ($\alpha = 1$) | **26.64** | **3.16** | 39.58 | **38.83** | **25.10** | 8.30 | **54.18** | **1.99** | 9.02 | **25.99** |
| | (3.89) | (0.77) | (0.77) | (2.38) | (4.39) | (8.12) | (3.46) | (0.28) | (3.37) | (11.48) |
| LLM-MAB ($\alpha = 2$) | 20.57 | 2.22 | 36.74 | 34.28 | 19.16 | **9.63** | 50.49 | 0.44 | **10.71** | 17.42 |
| | (2.99) | (1.68) | (2.20) | (1.49) | (1.18) | (9.35) | (4.08) | (1.00) | (5.35) | (17.98) |

**Sensitivity analysis of the exploration coefficient $\alpha$.** We further analyze the sensitivity of *LLM-MAB* to the exploration coefficient $\alpha$ in UCB, tested at $\alpha \in \{0.5, 1, 2\}$ (Table 5). All three settings consistently outperform the default solver, indicating that the framework is robust to this hyperparameter. Among them, $\alpha = 1$ provides the best overall trade-off, outperforming baselines on 8 out of 10 benchmarks and achieving the largest gains on hard instances such as `obj_3` and `all`. By contrast, $\alpha = 0.5$ favors exploitation and explores too little, resulting in faster convergence but is hard to yield the strongest performance. On the other hand, $\alpha = 2$ encourages aggressive exploration, which occasionally discovers stronger configurations (e.g., two benchmark wins), but its overhead in the early rounds often offsets these gains. These results suggest that moderate exploration ($\alpha = 1$) is most effective in practice, while also demonstrating that the overall framework remains stable across a wide range of $\alpha$ values.

**Sensitivity analysis of the configuration candidate set size $N$.** Table 9 in Appendix A.5 reports the performance of *LLM-MAB* under different numbers of configuration candidates, tested at $N \in \{1, 3, 5\}$. Note that $N = 1$ corresponds to *LLM-cold-start*, where a single configuration is selected without online adaptation. We observe that when $N = 5$, *LLM-MAB* achieves the most stable improvements and consistently outperforms default SCIP across all benchmarks, suggesting that having a sufficiently diverse candidate set is crucial to include at least one effective configuration. On the other hand, given the small size of the re-optimization benchmark (50 instances per problem series, of which only 45 are used for testing in our setting), larger values of $N$ are less practical: convergence becomes more difficult and each arm must still be explored at least once, leading to higher overhead and less reliable adaptation.

## 6 CONCLUSION

In this paper, we formulated the re-optimization problem for MILP by incorporating a finite history window of solver information, and we cast algorithm configuration as a joint problem of configuration-space reduction and adaptive online selection. Building on this formulation, we developed a lightweight two-stage method that first leverages LLMs to generate a compact set of candidate configurations and then applies bandit algorithms for online adaptation. Empirical results on the MIP Workshop 2023 re-optimization benchmarks demonstrate that our framework consistently improves solver performance over strong baselines, achieving substantial reductions in solving time without requiring offline training or validation data.

For future work, our framework can be extended by incorporating richer re-optimization techniques as configurable options, such as learning-based solution prediction or adaptive branching strategies. It is also promising to explore instance-wise configuration policies that exploit problem features more directly while retaining the lightweight online nature of our approach.

REPRODUCIBILITY STATEMENT

Our experiment setup (devices, solver settings, hyper-parameters, evaluation metrics, and baselines) is detailed in Section 5.1 and Appendix A.1, and the re-optimization benchmark is summarized in Section 5.1 and Appendix A.2. An anonymous code repository is provided in the supplemental materials to reproduce the experiment results.

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

# A APPENDIX

## A.1 EXPERIMENT SETUPS

**Algorithm configuration space.** In this work, we cast algorithm configuration as a joint selection over (i) solver-native parameters and (ii) optional re-optimization mechanisms. To keep the algorithm configuration space impactful yet tractable for MILP, we primarily target cutting-plane–related controls, which strongly influence LP bound quality, node counts, and overall time. As an optional re-optimization mechanism, we include Primal Hint (incumbent injection + light repair), which can provide high-quality starting solutions and thereby accelerate early pruning.

Our framework is solver-agnostic, and in this study, we restrict attention to features that are available in both SCIP and Gurobi. Consequently, we do not consider branching-information reuse here; investigating it within our framework is left for future work.

Table 6 and Table 7 provide the parameter lists for SCIP and Gurobi, respectively.

Table 6: List of SCIP parameters used in our generalized algorithm configuration. Values: 1 = on, 0 = off, respectively.

| Parameter | Description | Values |
|---|---|---|
| clique | Clique inequalities from conflict graphs | 0 / 1 |
| root_only | Restrict cuts to root node only | 0 / 1 |
| Primal Hint | Warm-start from prior solutions | 0 / 1 |
| cmir | Mixed-integer rounding cuts | 0 / 1 |
| aggregation | Flow-cover inequalities (aggregation) | 0 / 1 |
| mcf | Flow path cuts (multi-commodity flow) | 0 / 1 |
| impliedbounds | Implied bound cuts (bin-cont vars) | 0 / 1 |
| strongcg | Strong Chvátal–Gomory cuts | 0 / 1 |
| zerohalf | Zero-half inequalities | 0 / 1 |
| disjunction | Disjunctive cuts | 0 / 1 |
| convexprojection | Convex projection cuts (MINLP relaxations) | 0 / 1 |
| integerobjective | Objective integrality cuts | 0 / 1 |
| gomory | Gomory fractional cuts | 0 / 1 |
| cgmip | Chvátal–Gomory cuts | 0 / 1 |
| oddcycle | Odd cycle inequalities (graph-based) | 0 / 1 |
| rapidlearning | Rapid learning heuristic cuts | 0 / 1 |

**Implementation of Primal Hints.** The re-optimization dataset from the MIP Workshop 2023 Computational Competition on Re-optimization consists of problem series where different components vary across instances (objective function coefficients(obj), constraint right-hand sides(rhs), variable bounds(bnd), coefficients of the constraint matrix(mat). Changing obj does not affect feasibility, whereas changes to rhs/bnd/mat can render a previously feasible incumbent infeasible. Guided by this, we adopt the following hinting policy.

- obj series: we reuse the entire historical incumbent as the Primal Hint.
- rhs/bnd/mat series: we reuse only the integer part of the incumbent. For SCIP, we fix those integers and solve a continuous LP to repair the continuous variables; if a feasible completion is found, the full repaired solution is passed as the Primal Hint. For Gurobi, we provide the integer assignments as a partial MIP start and rely on its built-in repair mechanism to complete missing values. For bnd changes specifically, integer values are first clipped to the new bounds before repair.

**Experiment Setup Details of Baselines.** We compare against four baselines:

- *Default solvers (Default).* SCIP and Gurobi with default configurations.
- *Progressively Tuning (Tuning)* Patel (2024b). The winning solution of the MIP Workshop 2023 re-optimization competition. Each parameter (Provide hint or not, Use root node cuts

Table 7: List of Gurobi parameters used in our generalized algorithm configuration. Values: 2 = aggressive, 1 = on, 0 = off, respectively.

| Parameter | Description | Values |
|---|---|---|
| Primal Hint | Warm-start from prior solutions | 0 / 1 |
| CliqueCuts | Clique inequalities from conflict graphs | 0 / 1 / 2 |
| CoverCuts | Cover inequalities for knapsack sets | 0 / 1 / 2 |
| FlowCoverCuts | Flow-cover cuts for fixed-charge flows | 0 / 1 / 2 |
| FlowPathCuts | Path cuts in fixed-charge networks | 0 / 1 / 2 |
| GUBCoverCuts | Cover cuts under generalized upper bounds | 0 / 1 / 2 |
| ImpliedCuts | Implied bound inequalities (bin-cont vars) | 0 / 1 / 2 |
| InfProofCuts | Infeasibility proof inequalities | 0 / 1 / 2 |
| LiftProjectCuts | Lift-and-project cuts from disjunctions | 0 / 1 / 2 |
| MIRCuts | Mixed-integer rounding cuts | 0 / 1 / 2 |
| MixingCuts | Mixing inequalities (generalized MIR) | 0 / 1 / 2 |
| ModKCuts | Modular arithmetic divisibility cuts | 0 / 1 / 2 |
| NetworkCuts | Network structure cuts | 0 / 1 / 2 |
| RelaxLiftCuts | Relax-and-lift cuts | 0 / 1 / 2 |
| SubMIPCuts | SubMIP-based cuts | 0 / 1 / 2 |
| StrongCGCuts | Strong Chvátal–Gomory cuts | 0 / 1 / 2 |
| ZeroHalfCuts | Zero-half inequalities | 0 / 1 / 2 |
| ProjImpliedCuts | Projected implied bound inequalities | 0 / 1 / 2 |

or not, Use cuts at other nodes or not) is modeled as an independent bandit problem, with arms corresponding to its discrete values (on/off). The score of each value is computed by Upper Confidence Bound (UCB) Algorithm:

$$S_v = Q_v + \frac{C}{N_v}.$$

This score involves two parts: (i) The running average of the base score ($Q_v$), and (ii) The uncertainty of the score, i.e., the confidence bound $\frac{1}{N_v}$, where $N_v$ is the number of score updates the value has received. When a value is not explored enough, the confidence bound adds a higher number to the score to encourage more exploration for that value. $C$ is a weight constant that determines how fast the score converges and is fixed to $0.3$ as suggested by the paper. This baseline is implemented only for SCIP, as Gurobi does not expose root/non-root separation controls.

- *LLM with validation (LLM-validation)* Lawless et al. (2025). Following *LLM-MAB*, we prompt an LLM to generate 100 separator configurations for each re-optimization series, then apply $k$-medoids to condense them into $N = 5$ representatives. We evaluate each representative on a held-out validation set and select the configuration with the best average reward. This forms a strong baseline because it exploits additional validation instances to pick the best among LLM-generated candidates.

- *LLM-cold-start*. Again following *LLM-MAB*, we generate 100 separator configurations per series and cluster them with $k$-medoids, but we directly take the single medoid of the *largest* cluster without any validation as the chosen configuration for evaluation.

## A.2 RE-OPTIMIZATION BENCHMARKS

**Benchmark datasets.** Table 8 summarizes the re-optimization benchmarks used in our experiments, including the size, domain/source, and which model components vary within each sequence. Here, #Vars denotes the total number of variables (integer variables in parentheses), and #Cons denotes the number of constraints. An *X* indicates that the corresponding component changes across instances in the sequence. To analyze performance across different difficulty levels, we categorize the datasets into easy and hard groups. Easy datasets involve only a single varying (e.g., objective coefficients), and have relatively small problem sizes, with no more than $1,457$ integer variables. Hard datasets are characterized either by having at least two (up to four) varying components, by significantly larger problem sizes (up to $63,009$ integer variables), or by both.

Table 8: Re-optimization benchmark datasets: size, source domain, and varying components. #Vars: total number of variables (integer variables in parentheses). LO: variable lower bounds; UP: variable upper bounds; OBJ: objective coefficients; LHS: left-hand sides; RHS: right-hand sides; MAT: constraint matrix coefficients. *X* indicates that the corresponding component varies within the instance sequence.

| Dataset | #Vars | #Constrs | Domain/Source | LO | UP | OBJ | LHS | RHS | MAT |
|---------|-------|----------|---------------|----|----|-----|-----|-----|-----|
| *Easy datasets* | | | | | | | | | |
| bnd_2 | 1758 (1457) | 351 | MIPLIB'17 | X | X | – | – | – | – |
| bnd_3 | 1758 (1457) | 351 | MIPLIB'17 | X | X | – | – | – | – |
| rhs_2 | 1000 (500) | 1250 | Synthetic MILP | – | – | – | – | X | – |
| rhs_4 | 1000 (500) | 1250 | Synthetic MILP | – | – | – | X | X | – |
| obj_1 | 360 (360) | 55 | Stochastic multi 0–1 knapsack | – | – | X | – | – | – |
| *Hard datasets* | | | | | | | | | |
| rhs_3 | 63009 (63009) | 507 | MIPLIB'17 | – | – | – | X | X | – |
| obj_3 | 9599 (9599) | 27940 | UCI Machine Learning | – | – | X | – | – | – |
| rhs_obj | 90983 (60146) | 33438 | Hydro unit commitment | – | – | X | X | X | – |
| mat | 802 (500) | 531 | Vaccine allocation | – | – | – | – | – | X |
| all | 7973 (5186) | 3558 | Mixed synthetic benchmark | X | X | X | X | X | X |

After the table, we provide detailed descriptions of each dataset, including how instances were generated and which variations are introduced:

bnd_2: Based on `csched007` (MIPLIB 2017). Instances are generated via random fixings of 15%–25% of the binary variables, selected uniformly with respect to the original instance.

bnd_3: Also based on `csched007`. Instances are generated via random fixings of 5%–20% of the binary variables (uniform selection). This series is relatively harder than `bnd series 2` (as reflected by the time limits).

rhs_2: Based on a synthetic MILP dataset. Instances are generated by taking a convex combination of two different RHS vectors.

rhs_4: Also based on the synthetic MILP dataset as in `rhs series 2`, but using a different pair of RHS vectors for the convex combination.

obj_1: Based on a stochastic multiple binary knapsack dataset. We consider one scenario at a time, yielding a series in which roughly one third of the objective vector (corresponding to $y$-variables) varies across instances.

rhs_3: Based on `glass4` (MIPLIB). Instances are generated by perturbing nonnegative LHS and RHS components via a discrete uniform distribution by up to $\pm 70\%$ of their values.

obj_3: Derived from the UCI `MAGIC` dataset. Instances are subproblems from a column generation approach to decision trees; the final set stems from a public call for additional datasets.

rhs_obj: Based on a hydro unit commitment (HUC) MILP for a fixed valley. Varying inputs include electricity prices (objective), inflows, and initial/target reservoir volumes (constraint sides); most other data remain unchanged, making sequential re-optimization especially relevant.

`mat`: Based on an optimal vaccine allocation problem. Considering 500 scenarios at a time yields a series where the *constraint matrix of inequality constraints* varies across instances.

`mat_rhs_bnd_obj` `(all)`: Also HUC-based (as in `rhs_obj`), but here *every* input component (objective, bounds, sides, and matrix) may vary.

## A.3 LLM PROMPT EXAMPLE

Figures 2 and 3 provide an example LLM dialogue for algorithm configuration. The LLM input consists of (i) a meta prompt, (ii) a parameter description, and (iii) a problem description. The LLM output is a proposed configuration accompanied by brief explanations.

---

**Meta Prompt**

You are configuring for MILP re-optimization: a sequence of closely related instances derived from the same base model with small changes to objective function coefficients/variable bounds/constraint right-hand sides/coefficients of the constraint matrix. You need to configure the following solver parameters and global control options. Any configuration key not explicitly included in your final output will be set to 0 (disabled/default).

---

**Parameter Description Example: Primal Hint**

**Name:** Primal Hint

**What it is.** Provide an incumbent candidate solution before or during the solve (e.g., from a prior run or predictor). The solver can accept it directly or attempt to repair it.

**How it maps to solvers.** When objective coefficients change, the entire historical incumbent can be reused as the Primal Hint. When constraint right-hand sides, variable bounds, or matrix coefficients change, only the integer part of the incumbent is reused.    – In SCIP, these integer values are fixed, and a continuous LP repair is attempted for the remaining variables; if feasible, the repaired solution is passed as the Primal Hint. – In Gurobi, the integer assignments are provided as a partial MIP start, with its built-in repair mechanism completing the solution. For bound changes, integer values are clipped to the new bounds before repair.

**When it helps.** Primal hints can significantly reduce time to derive a feasible solution at the early stage and improve pruning efficiency when the hint is of high quality and structurally consistent across instances. They are especially effective when only objective coefficients values vary, since the incumbent remains mostly compatible. In contrast, when major structural changes occur (e.g., large-scale bound shifts or constraint matrix modifications), reusing incumbents is less reliable and may even increase search tree size if the hint is of poor quality.

---

**LLM Output Example: Proposed Configuration**

**Setting levels:** 0=*disable*, 1=*use*, 2=*aggressive*.

| Parameter | Level | Explanation |
|---|---|---|
| primalhint | 1 | Given the series of 50 instances with common models, enabling primal hints allows warm starting from previous solutions, potentially speeding up reoptimization. |
| root_only | 1 | With primal hints, strong root relaxations can suffice, saving time without weakening the initial solution quality. Larger search tree risk is mitigated by more frequent feasible primal hints. |
| clique | 1 | Useful for binary decision variables, especially with potential binary onflicts in multi-dimensional knapsack constraints. |
| impliedbounds | 1 | Exploiting logical bounds between binary variables in the problem, helping improve LP relaxation efficiency given binary structure. |
| strongcg | 1 | With pure binary and integer covering or tight knapsack constraints, these strengthen LP relaxation and combinatorial logic. |

---

Figure 2: LLM dialogue example for algorithm configuration. From top to bottom: (i) Meta prompt and (ii) Parameter Description Example. These two boxes are the materials provided to the LLM. The final box shows the LLM output: a concrete configuration with multi-level settings.

---

**Problem Description: Stochastic Multiple Binary Knapsack (obj series 1)**

**Name:** Stochastic Multiple Binary Knapsack (single-scenario instantiation)

**What it is.** Choose binary items in two stages. First-stage selections $x, z$ must satisfy multi-dimensional covering constraints; after the scenario is revealed, a second-stage recourse $y$ covers residual requirements.

The objective is linear, with scenario-dependent coefficients on $y$. This series varies (parts of) the objective vector across instances.

**Data and notation.** *Decision variables: $x \in \{0, 1\}^n$, $z \in \{0, 1\}^n$ (first-stage), $y \in \{0, 1\}^n$ (second-stage/recourse). Costs: $c, d \in \mathbb{R}_{\geq 0}^n$ for $x, z$; scenario-specific $q \in \mathbb{R}_{\geq 0}^n$ for $y$. Constraints: covering matrices $A, C, W, T$ and right-hand sides $b, h$ with compatible dimensions.*

**Dataset metadata.** Series size: 50 related instances (common model, varying data). Number of variables: 360 (all binary). Number of constraints: 55.

**Objective.** Minimize total cost of first-stage decisions plus recourse cost:

$$\min_{x, z, y} \; c^\top x + d^\top z + q^\top y.$$

**Integer programming formulation.**

Minimize $\quad c^\top x + d^\top z + q^\top y$

Subject to $\quad Ax + Cz \;\geq\; b \qquad\qquad\qquad$ (multi-dimensional covering)

$\qquad\qquad\quad Wy \;\geq\; h - Tx \qquad\qquad\quad$ (recourse covers residual demand)

**Constraints summary.**

- **Covering (multiple knapsacks):** $Ax + Cz \geq b$.
- **Recourse covering / linking:** $Wy \geq h - Tx$.

- **Binary-only:** all decision variables are binary.
- *Optional stochastic view:* in the general two-stage model, $q$ is scenario-dependent and one may minimize $\sum_{\omega \in \Omega} p_\omega \, Q_\omega(x)$; here a single scenario is fixed per instance.

**Re-optimization variant (obj1).** Varying component: the objective coefficients (typically a subset associated with $y$) vary across instances; feasibility structure is preserved.

Figure 3: LLM dialogue example of problem description.

## A.4 MAB ALGORITHM IMPLEMENTATION

For completeness, we provide in Algorithm 1 the pseudocode of our bandit-based online configuration selection procedure in (*LLM-MAB*). The algorithm maintains empirical mean rewards $\hat{\mu}_s$ and counts $n_s$ for each candidate configuration $s \in \mathcal{S}'$. At each step $t$, given an instance $P_t$, if some configurations have not yet been explored, they are selected to ensure every arm is tried at least once. Otherwise, the next configuration is chosen according to the UCB1 score with exploration coefficient $\alpha$. The solver is then run on the current re-optimization instance $P_t$, optionally reusing primal hints from the last $\tau$ instances. The solving time is converted into a negative reward and used to update statistics online. This procedure adaptively balances exploration of different configurations with exploitation of the most promising ones.

---

**Algorithm 1:** LLM-MAB: Online Configuration Selection via UCB1

---

**Input** : Exploration coefficient $\alpha$;
        Candidate configurations $\mathcal{S}' = \{s_1, \ldots, s_N\}$;
        Sequential instance stream $\{P_t\}_{t=1}^{T}$;
        Historical window size $\tau$ (for optional reuse of primal hints).
**Output:** Per-instance record of chosen configuration and solving time.

**Initialization:** For each $s \in \mathcal{S}'$, set $\hat{\mu}_s(0) \leftarrow 0$, $n_s(0) \leftarrow 0$.
**for** $t \leftarrow 1$ **to** $T$ **do**
  **if** $\exists s : n_s(t-1) = 0$ **then**
      Select such untried configuration $s_t = s$.       `// ensure each tried once`
  **else**
      Compute UCB1 score $\mathrm{U}_s(t) = \hat{\mu}_s(t-1) + \alpha\sqrt{\frac{2\ln t}{n_s(t-1)}}$ for each $s$.
      Select $s_t = \arg\max_{s \in \mathcal{S}'} \mathrm{U}_s(t)$.
  Run solver on $P_t$ with configuration $s_t$ (and hints from last $\tau$ instances if enabled).
  Observe solving time $z_t$, set reward $r_t = -z_t$.
  Update statistics: $n_{s_t}(t) \leftarrow n_{s_t}(t-1) + 1$,
  $\hat{\mu}_{s_t}(t) \leftarrow \hat{\mu}_{s_t}(t-1) + \frac{r_t - \hat{\mu}_{s_t}(t-1)}{n_{s_t}(t)}$.

---

## A.5 Additional Experiment Results

**Sensitivity analysis of the configuration candidate set size.** Table 9 reports the detailed results of *LLM-MAB* with different candidate set sizes $N \in \{1, 3, 5\}$ across all benchmarks. As discussed in Section 5.2, increasing $N$ to 5 provides more stable performance and ensures consistent improvements over the default solver.

Table 9: Sensitivity analysis of the configuration candidate set size. We report relative solving time improvement (%) over Default SCIP across 10 re-optimization benchmarks. (Higher is better.)

| Method | bnd_2 | bnd_3 | rhs_2 | rhs_4 | obj_1 | rhs_3 | obj_3 | rhs_obj | mat | all |
|---|---|---|---|---|---|---|---|---|---|---|
| LLM-MAB ($N = 1$) | 6.48 | -0.79 | **41.42** | 37.76 | 24.16 | 6.68 | 53.60 | **2.72** | 8.42 | 19.35 |
| | (2.62) | (1.77) | (1.56) | (0.67) | (1.80) | (17.52) | (6.38) | (0.47) | (2.99) | (18.92) |
| LLM-MAB ($N = 3$) | 11.06 | **13.39** | 38.45 | 32.68 | 20.92 | 1.57 | 47.07 | 0.88 | -0.84 | 10.36 |
| | (12.79) | (3.01) | (2.17) | (1.04) | (2.14) | (12.49) | (4.54) | (0.32) | (3.16) | (18.60) |
| LLM-MAB ($N = 5$) | **26.64** | 3.16 | 39.58 | **38.83** | 25.10 | 8.30 | **54.18** | 1.99 | **9.02** | **25.99** |
| | (3.89) | (0.77) | (0.77) | (2.38) | (4.39) | (8.12) | (3.46) | (0.28) | (3.37) | (11.48) |

**Additional experiments with extended time limit.** In preliminary experiments, we observed that within each re-optimization sequence, the default solver's solving times varied significantly across instances. To prevent a few particularly hard instances from dominating the aggregate metrics, we followed prior work Lawless et al. (2025) and imposed a global time limit of 400 seconds per instance. However, on certain datasets this budget proved insufficient, as many instances could not reach optimality within the limit. To ensure a fairer evaluation, we repeated the experiments with an extended time limit of 600 seconds. The corresponding results are reported in Table 10.

We observe that *LLM-MAB* still achieves consistent improvements over Default SCIP. In particular, substantial gains are observed on obj_1 (+33.42%) and mat (+12.29%), while on rhs_obj the performance is similar to the default. These results confirm that our approach remains effective under more relaxed computational budgets.

Table 10: Additional experiments with extended time limit. We report solving time (s) and relative improvement (%) over Default SCIP. Lower solving time and higher relative improvement are better.

| Method | bnd_3 | rhs_3 | obj_1 | rhs_obj | mat |
|---|---|---|---|---|---|
| Default | 493.16 (4.42) | 434.04 (32.60) | 409.32 (1.42) | 570.95 (5.20) | 504.33 (4.97) |
| LLM-MAB | **462.32** (3.21) | **394.25** (29.41) | **272.55** (7.95) | **569.46** (6.82) | **442.21** (25.53) |
| Improv. | 6.24 | 8.51 | 33.42 | 0.25 | 12.29 |

### A.6 The Use of Large Language Models

We used a large language model solely for language polishing and LaTeXformatting, such as grammar correction, stylistic editing, and minor reformatting of tables and figures. The LLM was not used to design experiments, generate results, analyze data, or draw conclusions. All technical content, algorithms, and empirical findings were authored and verified by the authors.

