# OpenReview forum: "Online Algorithm Configuration for MILP Re-Optimization with LLM Guidance"
_ICLR.cc/2026/Conference — Submitted to ICLR 2026_

### Official Review · Reviewer_joEs · 2025-10-27

**Soundness:** 2
**Presentation:** 2
**Contribution:** 2
**Rating:** 2
**Confidence:** 4

**Summary:**

This paper presents a two-stage framework for algorithm configuration in the context of MILP re-optimization. The method first leverages LLMs to generate a compact portfolio of candidate configurations, reducing a combinatorially large search space. It then employs a Multi-Armed Bandit algorithm for online adaptive selection from this portfolio across a sequence of related problem instances. The empirical evaluation on established benchmarks demonstrates performance improvements over default solver settings and strong baselines, achieving up to a 54.18% reduction in solving time without requiring offline training data.

**Strengths:**

1. The integration of LLMs for zero-shot algorithm configuration space reduction is highly innovative and represents a promising new direction for automating solver tuning.

2. The paper effectively formulates the re-optimization problem as a joint configuration of both built-in solver parameters and re-optimization techniques (like primal hints), which is a more holistic approach than tuning parameters in isolation.

**Weaknesses:**

1. The method heavily relies on the zero-shot capability of LLMs to generate high-quality configurations. While over-generation and clustering mitigate this, there is no theoretical foundation or guarantee for the performance or consistency of the LLM-generated portfolio. The approach is sensitive to prompt design, yet the paper lacks a systematic analysis of prompt robustness or variability.

2. The configuration space is restricted primarily to cutting-plane parameters and the primal hint mechanism, as shown in Tables 6 and 7. This overlooks other critical solver components such as branching rules, heuristics, and conflict analysis, which are known to have a profound impact on solver performance. This limitation may prevent the framework from achieving its full potential, especially on complex instances where a more holistic configuration is required.

3. The exploration coefficient (α) in the UCB algorithm is a critical hyperparameter that requires manual setting. The paper fixes α=1 based on sensitivity analysis but does not propose or implement an adaptive mechanism to adjust it during the online process.

4. The primary evaluation metric is solving time. While important, this does not provide a complete picture. Other crucial metrics like the primal-dual integral or optimality gap are not reported, which could reveal different performance characteristics, especially for instances not solved to optimality. Furthermore, even with an extended time limit of 600 seconds, the improvement on the most challenging dataset (`rhs_obj`) is minimal (0.25%), suggesting the method's effectiveness may be limited for particularly difficult re-optimization problems.

5. Generating 100 configurations via an LLM (e.g., using a commercial API) and performing clustering for each problem series incurs significant computational and potential financial cost.

6. The paper does not provide an analysis or theoretical insight into *how* or *why* the LLM arrives at certain configurations from the problem description.

**Questions:**

1. Have you conducted any analysis on the robustness of your method to variations in the LLM prompt?

2. What was the rationale behind limiting the configuration space to cutting planes and primal hints?

3. Could you comment on the performance in terms of optimality gaps for instances that hit the time limit? Would including metrics like the primal-dual integral change the relative ranking of the methods?

4. How significant is the computational overhead of the LLM generation and clustering phase compared to the total solving time?

---

### Official Review · Reviewer_SNKv · 2025-10-27

**Soundness:** 1
**Presentation:** 2
**Contribution:** 1
**Rating:** 2
**Confidence:** 5

**Summary:**

This paper focuses on the "re-optimization" setting for solving mixed-integer linear programs (MILP). The basic idea is that there is a sequence of problems with identical sizes/structures, but different coefficients/internal properties. The parameters of the MILP should be adjusted for each instance that is solved to minimize the runtime needed to solve the instance. Experiments are conducted along the lines of the MIP re-optimization challenge that was recently conducted.

**Strengths:**

1. The topic is simultaneously highly relevant for industry applications and methodologically interesting.
2. The MIP workshop creates a relatively good setting for testing out the proposed approach.

**Weaknesses:**

1. The paper ignores the algorithm configuration literature and proposes an LLM-based method without ever examining whether this is a good idea. There are many techniques already available for online algorithm configuration, see the survey by Schede et al. (2022) in JAIR. There is also work on AC landscapes, particularly on analyzing the parameters, see, e.g., Pushak and Hoos (2018) in PPSN. Simply put, the value add of the LLM is not investigated over existing techniques. And, intuitively, I have no idea why an LLM should be able to determine from analyzing descriptions of parameters which parameters are important or are not. This determination should be made through an analysis of algorithm performance.
2. The paper investigates a restricted re-optimization setting. I ask the authors to correct me if I am wrong, but the basic assumption here seems to be that A, c, and b do not change in size between problems. Under this assumption, we cannot even add an extra customer to a routing problem when resolving, which severely limits the applicability of the problem setting.

Let me note that I consider these such fatal flaws in the work that I did not read the rest in detail.

I also note that I found the MILP formulation in (1) odd: it is correct, but I wonder why standard form is insufficient?

**Questions:**

1. Why use an LLM rather than existing statistical methods to determine important parameters?
2. Why use an LLM at all in this AC process? What benefit does it provide?
3. Why are existing AC methods ignored?

---

### Official Review · Reviewer_2hya · 2025-10-29

**Soundness:** 2
**Presentation:** 3
**Contribution:** 2
**Rating:** 2
**Confidence:** 4

**Summary:**

This paper addresses MILP re-optimization by proposing a two-stage framework: first using LLMs to generate a reduced set of solver parameter configurations, then applying multi-armed bandit algorithms to adaptively select among these configurations online. The authors evaluate their method on 10 benchmark series from the MIP Workshop 2023, each containing 50 sequential instances with varying components (objective coefficients, constraints, bounds, etc.). The experimental results show solving time improvements ranging from 1-54% compared to default SCIP and Gurobi configurations, with stronger gains on harder instances. The method combines both built-in solver parameters (primarily cutting plane controls) and re-optimization mechanisms (warm starts) in a unified configuration space. The ablation studies indicate that both the general parameter tuning beyond just separators and the online MAB selection contribute to performance.

**Strengths:**

1. The progression from problem formulation to methodology to experiments follows a natural structure, making it easy to understand the authors' approach and contributions.
2. Unlike many algorithm configuration methods that need extensive validation sets or supervised learning, this approach works directly on the sequential instances without requiring additional data, making it more applicable to real-world re-optimization scenarios where instances are limited.
3. The framework successfully works with both SCIP and Gurobi solvers despite their different parameter spaces and internal architectures, suggesting the approach can generalize across various commercial and open-source MILP solver implementations.

**Weaknesses:**

First, the methodology pipeline lacks great novelty - using LLMs for configuration space reduction involves only straightforward prompting without any sophisticated techniques, and the paper fails to compare against other configuration reduction methods like heuristic-based space reduction methods that could serve as baselines (Li et al., 2023 on separator configuration). Additionally, using MAB for online configuration selection has been explored before (as acknowledged by their comparison with Patel 2024b), making the overall pipeline appear incremental rather than innovative.

Second, the experimental evaluation is limited to only 10 problem series from a single benchmark suite, with some series differing only in which components vary (e.g., bnd_2 vs bnd_3, rhs_2 vs rhs_4). The solving times are relatively small even for "hard" instances (many under 400 seconds), and the paper acknowledges that several instances couldn't reach optimality within the time limit, raising questions about whether these benchmarks adequately represent real-world MILP re-optimization challenges that practitioners face.

Third, the performance improvements over strong baselines are often marginal - while the paper claims up to 54% improvement, the comparison with LLM-coldstart (which simply selects one configuration without any online adaptation) shows differences of less than 2% on many benchmarks, and on some datasets like rhs_2, LLM-coldstart actually outperforms the proposed LLM-MAB method. This suggests the benefit of the bandit-based online selection component may not justify its additional complexity.

**Questions:**

Besides the concerns in the weakness part, I also have the following questions:

1. The authors use MAB without contextual information, but in the evaluation stage, the problem is changing and why MAB is still valid for usage? Contextual bandit that incorporates instance features (problem size changes, which components are varying, magnitude of changes) would be more theoretically sound for this non-stationary environment where configuration effectiveness depends on the current instance properties.

---

### Official Review · Reviewer_S2sH · 2025-11-01

**Soundness:** 2
**Presentation:** 2
**Contribution:** 1
**Rating:** 2
**Confidence:** 4

**Summary:**

The paper observes that near-optimal solver parameters tend to be similar across different instances in MILP reoptimization, while traditional learning-based methods require a large amount of training data and cannot perform online algorithm configuration. The paper proposes using an LLM combined with the Multi-Armed Bandit UCB algorithm to automatically perform online algorithm configuration.

**Strengths:**

1. The paper defines a MILP reoptimization problem that has practical relevance.
2. The proposed LLM-MAB framework performs strongly on the MILP reoptimization benchmark, outperforming the baselines.

**Weaknesses:**

1. By ICLR standards, the paper lacks sufficient technical novelty. The MILP reoptimization problem is valuable, but the proposed LLM-MAB approach appears to be a straightforward combination of existing components.
2. In the Multi-Armed Bandit section (Section 4.2), the paper does not justify the choice of the UCB algorithm and should consider comparisons with other MAB algorithms.

**Questions:**

1. As an online reoptimization algorithm for a set of MILP problems, I suggest the authors compare LLM-MAB with a Bayesian optimization approach over hyperparameters, where the optimization objective is to minimize the solving time.
2. The performance of this method could be heavily influenced by the initial parameter configurations provided by the LLM. Therefore, the authors should investigate how well the LLM generates these configurations for different MILP problems. Does the LLM generally provide good candidate configurations? How much do these candidate configurations vary across problems? How does using different LLMs affect these results?

---

### Meta-Review · Area_Chair_9Uor · 2025-12-20

**Summary:**

The paper addresses a practical and relevant problem: Mixed-Integer Linear Programming (MILP) re-optimization, where a sequence of similar problem instances must be solved efficiently. The authors propose a two-stage framework, LLM-MAB, which uses Large Language Models (LLMs) to reduce the massive configuration space of solver parameters into a compact portfolio, followed by a Multi-Armed Bandit (MAB) algorithm (specifically UCB1) to adaptively select the best configuration online.

While the problem setting is valuable and the empirical results show improvements (up to 54.18% solving time reduction) , the submission faces significant hurdles regarding technical novelty and methodological depth. The core concerns informing this decision are:

- Incremental Innovation: The pipeline is a straightforward combination of existing tools (LLMs for prompting, MAB for selection) without a deep integration or novel theoretical insight.
- Lack of Competitive Baselines: The paper fails to compare against established Algorithm Configuration (AC) literature or more sophisticated statistical methods for parameter importance, relying instead on a "black-box" LLM approach.
- Limited Scope and Robustness: The configuration space is heavily restricted (mostly separators), and the method's sensitivity to prompt engineering—a known volatility in LLM applications—is not systematically addressed.

**Reviewer Concerns:**

I didn't see authors' response, therefore no concerns from reviewers have been address.

Outstanding concerns include:

- Technical Novelty: The primary weakness remains that the work appears as a "straightforward combination of existing components". The use of LLMs for simple zero-shot prompting lacks the sophistication expected for a top-tier machine learning conference.
- Omission of AC Literature: The paper essentially ignores decades of research in online algorithm configuration (e.g., Schede et al., 2022). Without comparing LLM-based reduction to statistical landscape analysis, the "value add" of the LLM remains unproven.
- Benchmarking Quality: The problem series are relatively small (50 instances), and solving times are often low, making the overhead of LLM generation and early MAB exploration a significant portion of the total cost.
- Restrictive Setting: The assumption that constraint matrices ($A$), objective vectors ($c$), and right-hand sides ($b$) do not change in size limits real-world applicability (e.g., adding a new customer to a routing problem).

**Reviewer Scores:**

- S2sH (Initial: 2): No change (2).
- 2hya (Initial: 2): Stay at 2.
- SNKV (Initial: 2): No change (2).
- joEs (Initial: 2): Stay at 2.

---

### Decision · Program_Chairs · 2026-01-26

Reject